# Tesla valves and capillary structures-activated thermal regulator

Wenming Li[1], Siyan Yang[2,3], Yongping Chen [1,4] ✉, Chen Li[5] & Zuankai Wang [2] ✉

Two-phase (liquid, vapor) flow in confined spaces is fundamentally interesting and practically important in many practical applications such as thermal management, offering the potential to impart high thermal transport performance owing to high surface-to-volume ratio and latent heat released during liquid/vapor phase transition. However, the associated physical size effect, in coupling with the striking contrast in specific volume between liquid and vapor phases, also leads to the onset of unwanted vapor backflow and chaotic two-phase flow patterns, which seriously deteriorates the practical thermal transport performances. Here, we develop a thermal regulator consisting of classical Tesla valves and engineered capillary structures, which can switch its working states and boost its heat transfer coefficient and critical heat flux in its "switched-on" state. We demonstrate that the Tesla valves and the capillary structures serve to eliminate vapor backflow and promote liquid flow along the sidewalls of both Tesla valves and main channels, respectively, which synergistically enable the thermal regulator to self-adapt to varying working conditions by rectifying the chaotic two-phase flow into an ordered and directional flow. We envision that revisiting century-old design can promote the development of next generation cooling devices towards switchable and very high heat transfer performances for power electronic devices.

Miniaturization of electronic devices demands the urgent development of highly efficient cooling approaches to timely dissipate the ever-increasing heat[1–3]. Because of excellent heat dissipation ability and availability of coolant, two-phase (vapor and liquid) heat transfer such as spraying cooling[4,5], pool cooling[1,6–8] and flow boiling[9–11], has been extensively explored by passively tailoring surface wettability, topography, or actively applying external electrical and magnetic fields, etc. In spite of extensive progress, it remains challenging to regulate two-phase mass and heat transport in the confined space owing to manifesting multiple spatial and temporal scales as well as

dramatic differences in the thermophysical properties of liquid and vapor phases.

As an extension of directional single-phase flow that is widely manifested in living organisms[12–16] and engineered devices[3,17], the preferential two-phase flow in open space has been proposed by designing cellular-fluidic devices[15] that leverage unit-cell-based, three-dimensional structures. Recently, by separating the vapor generation and evacuation in the vapor channel and liquid refilling in different pathways, a directional two-phase flow in open space was achieved, which sustains effective thermal cooling even at very high

[1]Key Laboratory of Energy Thermal Conversion and Control of Ministry of Education, School of Energy and Environment, Southeast University, Nanjing 210096, PR China. [2]Department of Mechanical Engineering, Hong Kong Polytechnic University, Hong Kong, PR China. [3]Department of Mechanical Engineering, City University of Hong Kong, Hong Kong, PR China. [4]Jiangsu Key Laboratory of Micro and Nano Heat Fluid Flow Technology and Energy Application, School of Environmental Science and Engineering, Suzhou University of Science and Technology, Suzhou 215009, PR China. [5]Department of Mechanical Engineering, University of South Carolina, Columbia, SC, USA. ✉e-mail: ypchen@seu.edu.cn; zk.wang@polyu.edu.hk

temperatures[18]. Moving from open space to closed channels, however, achieving directional two-phase flow encounters more challenges. First, the two-phase flows in confined spaces becomes more chaotic, random and uncontrollable[19,20]. Although an ordered single-annular flow can be achieved by decorating micro-pinfin fences along the channel sidewalls[21], it still suffers from uncontrolled vapor backward flow and deteriorated boiling heat transfer[22,23]. Second, a daunting barrier perplexing conventional two-phase heat transfer is the mutual exclusiveness in HTC and CHF. An increase in the heat flux will inevitably lead to a significant drop in HTC, which poses unforeseen temperature rise and even thermal crisis, especially in constrained space. Hence, new self-adaptive cooling technologies for various working conditions can ensure high reliability and high security.

Inspired by Tesla valves, invented by Nikola Tesla in 1920[24], here we design a thermal regulator, as schematically illustrated and displayed in Fig. 1a, that is capable of switching its working states through transforming the chaotic two-phase flow into a directional transport to achieve dramatic boost in heat transfer performances. The key elements of thermal regulator include Tesla valves (Fig. 1b) and regularly patterned pillar arrays (Fig. 1c) decorated on the sidewalls of both Tesla valves and main channels. The Tesla valves are designed to consist of Tesla island and Tesla bend, with an aim to generate substantial flow resistance difference two flow directions. The patterned pillar arrays are designed to trap a preferred and continuous liquid layer for effective thin film evaporation[6,25,26]. We expect that the fusion of these two key structural elements will lead to synergistic cooperation in inducing directional two-phase flow and highly efficient thermal cooling.

## Results and discussion

### Design and characterization of thermal regulator

We implemented standard photolithography and deep reactive ion etching (DRIE) to fabricate the Tesla valves as well as patterned pillar arrays, followed by the anodic bonding with the Pyrex glass to construct a fully integrated thermal regulator (characterized with hydraulic diameter $D_h \approx 162\,\mu m$). Figure 1a also presents a closed-view of the as-fabricated thermal regulator as well as the distinctive liquid and vapor flow pathways. The main channel, pillars, Tesla island and Tesla bend have an identical height of 250 $\mu m$, while the width of main channel is twice of the Tesla bend (100 $\mu m$). Note that the DRIE process leads to a roughness of ~350 nm on sidewalls of pillars with a local water contact angle of $\theta \approx 0°$. In contrast, it is about 45° on the bottom surface. Thus, the capillary pressure within the meniscus is calculated as $\Delta P_c = \frac{4\gamma\cos\theta}{d}$, where $d$ is the gap between pillars and sidewalls and $\gamma$ is the surface tension, for which the superhydrophilic pillar arrays are treated as capillary structures. In our design, the capillary pressure is ~19.2 kPa, over 13 times higher than the plain-wall microchannel with the same channel width. For comparison, we also fabricated two control samples. The first one is Tesla channels ($D_h \approx 222\,\mu m$; Fig. 1d), consisting of channels with Tesla valves alone. The other is plain-wall microchannels without the presence of typical Tesla valves (Supplementary Fig. 1).

### Roles of Tesla valves and capillary structures

To demonstrate the distinctive effects of Tesla valves and capillary structures, we first focused on Tesla channel and measured the single-phase pressure drops in forward and backward directions to decouple the influence of capillary structures. Here, we define the pressure drop

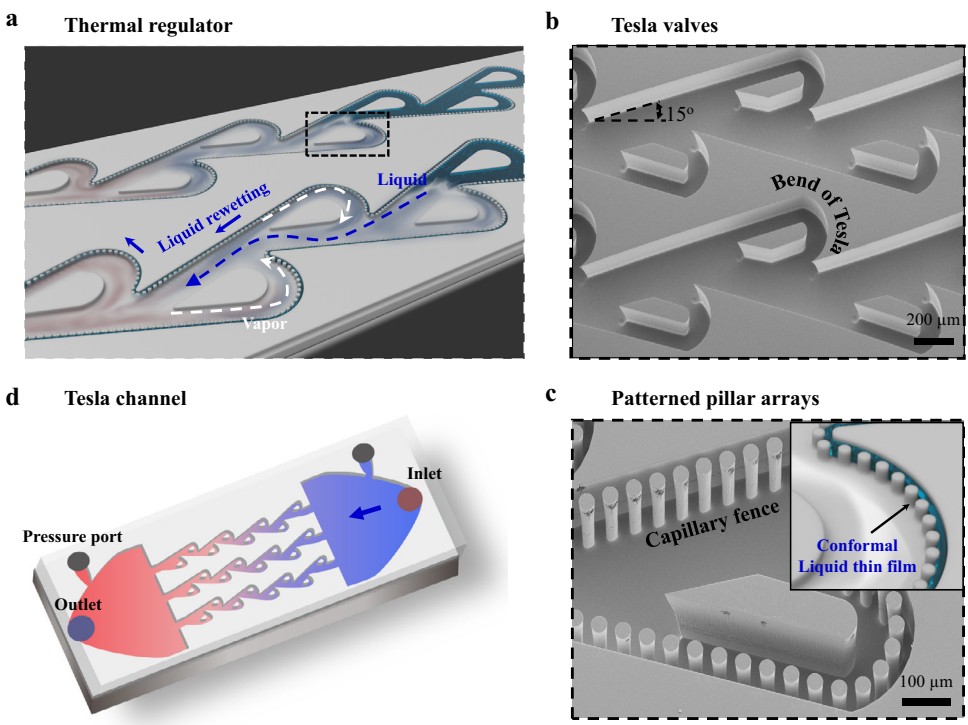

**Fig. 1 | Design and characterization of thermal regulator. a** Schematic diagram of thermal regulator, which consists of two key elements, Tesla valves and regularly patterned pillar arrays. Conformal thin liquid film (dashed blue arrow) is formed between patterned pillar arrays and sidewall of channel, and the severe vapor backward flow (dashed white arrow) can be rectified by Tesla valves to achieve directional two-phase flow. **b** The enlarged scanning electron microscopic (SEM) image illustrates the configuration of the as-fabricated Tesla valves comprised of the Tesla island and Tesla bend, which have an identical height with the main channel of 250 $\mu m$, while the width of Tesla valves is twice of the Tesla bend (100 $\mu m$). **c** SEM image of the patterned pillar arrays decorated on the sidewalls of Tesla valves and the main channel. The schematic diagram shows the liquid boundary layer (inset). The diameter, spacing of pillars is 30 $\mu m$ and 10 $\mu m$, respectively, and the gap between sidewall and pillars is 15 $\mu m$. **d**, Schematic view of typical Tesla channels for controlling two-phase (liquid and vapor) flow in one direction.

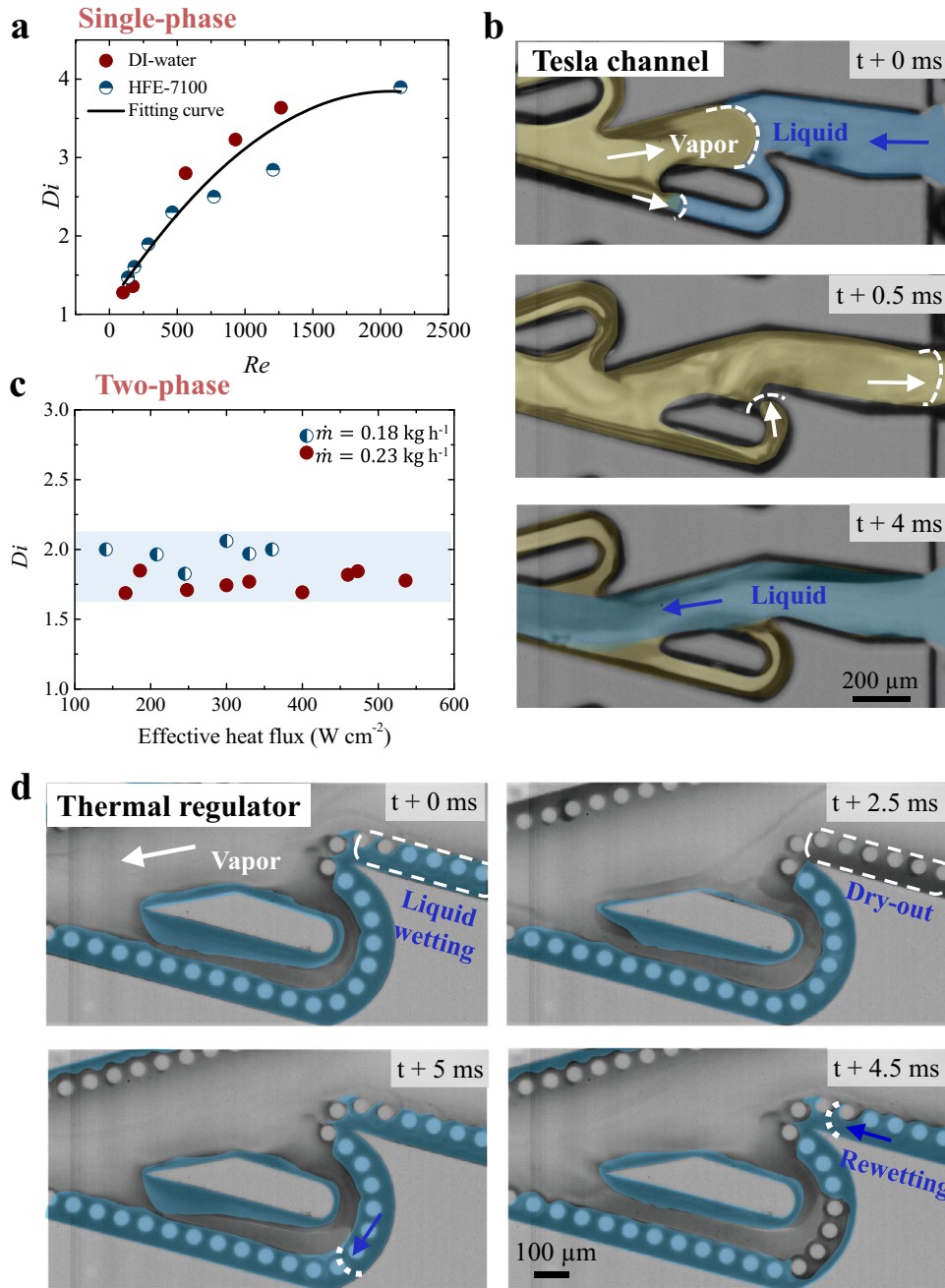

**Fig. 2 | Roles of Tesla valves and capillary structures. a** The variation of flow diodicity ($Di$) with Reynolds number ($Re$) characterized by the single-phase flow of DI-water and HFE-7100. $Di$ is defined as the ratio of total measured pressure drop in the backward direction and forward direction ($D_i = \frac{\Delta p_r}{\Delta p_f}$). **b** Time sequential optical snapshots showing the rectification of vapor backward flow within the Tesla valves (the vapor is recolored with yellow and liquid is recolored in blue, Supplementary Movie 1). $\dot{m} = 0.18$ kg h⁻¹ and heat flux is 354 W cm⁻². **c** The two-phase flow diodicity under different working conditions. **d** In the thermal regulator, the design of superhydrophilic pillar arrays leads to a repeated liquid wetting and rewetting (recolored in blue), avoiding dry-out along the sidewall. $\dot{m} = 0.18$ kg h⁻¹ and heat flux is 500 W cm⁻² (Supplementary Movie 3). Source data are provided as a Source Data file.

in the backward direction relative to that in the forward direction as the flow diodicity. The pressure loss $\Delta p$ in Tesla bend is expressed by $\Delta p = \frac{1}{2}f_s\rho u^2 \frac{\pi R_b}{D}\frac{\theta}{180°} + \frac{1}{2}k_b\rho u^2$ (detailed in "Methods" section). As shown in Fig. 2a, the diodicity of Tesla channel based on both DI-water and HFE-7100 rises as Reynolds number ($Re$) increases, reminiscence of a diode-like flow. Note that under similar $Re$, the diodicity endowed by the Tesla valves is about two-fold higher than that reported by Q. Nguyen et al.[27].

Moving to two-phase flow, we find that Tesla channel exhibits different two-phase transport behaviors as shown in Fig. 2b. Focusing on one Tesla unit reveals that water mainly flows in the forward

direction whereas the newly generated vapor also flows back towards the inlet. When the vapor reaches the Tesla island, the main channel is separated into two pathways. Notably, one-third of vapor selects to flow along the narrow, curved Tesla bend, which serves to steer the flow in the opposite direction with attenuated flow momentum. Accordingly, the chaotic two-phase flow is rectified into a directional and ordered profile (Supplementary Movie 1 and Supplementary Fig. 2). Such a rectification of vapor flow is further amplified by numerous Tesla units evenly distributed inside the main channel, which collectively result in the stagnation of vapor backflow and subsequent liquid reflushing into the channels. During this periodic

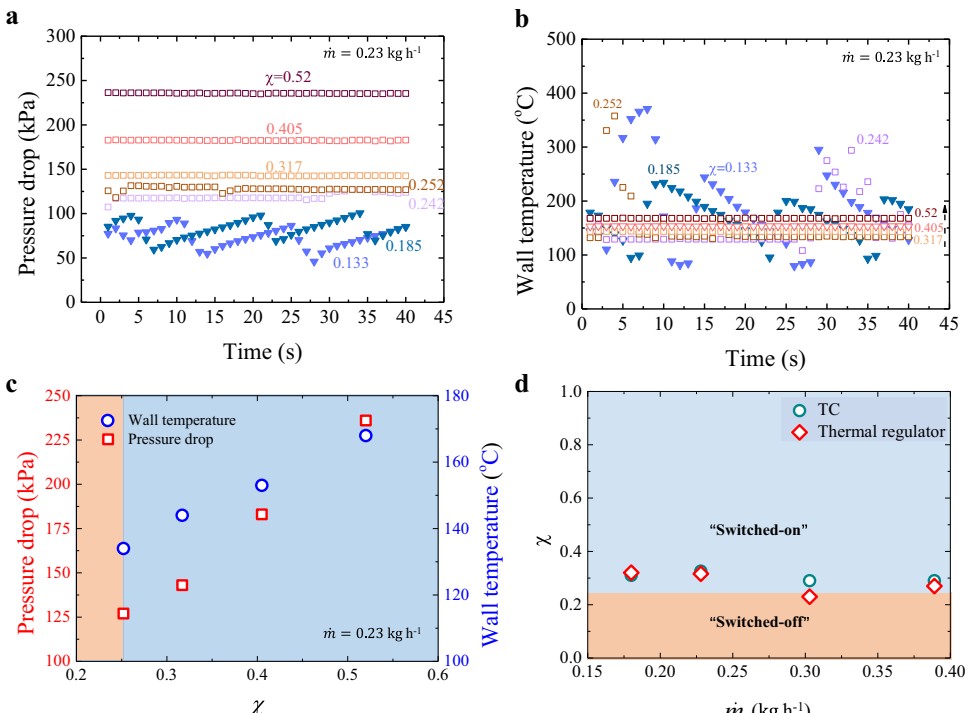

**Fig. 3 | Determination of the switching criterion for thermal regulator in the forward direction. a, b** Plot of the time-dependent variation of pressure drop and wall temperature under varying $\chi$ ranging from 0.133 to 0.52. When $\chi < 0.25$, both pressure drop and wall temperature fluctuate periodically corresponding to the chaotic two-phase flow (Supplementary Fig. 6), indicating the "off" state of the thermal regulator. In contrast, when $\chi > 0.25$, the pressure drop and wall temperature remain unchangeable, switching to an "on" state, in which two-phase backflow is completely rectified (Supplementary Fig. 7). **c** In the switched-on state, $\chi$ directly regulates the pressure drop and wall temperature (in orange zone, $\chi < 0.25$, whereas in blue zone, $\chi > 0.25$). Here $\dot{m} = 0.23$ kg h⁻¹. **d,** A critical $\chi$ of 0.25 indicates the switching criterion of thermal regulator and Tesla channel. The error bars for $\chi$ are -±1%. Source data are provided as a Source Data file.

process, the vapor evacuation mainly occurs in the forward direction, though a localized backward flow is also manifested in a limited duration (Supplementary Movie 2). In striking contrast, plain wall microchannels without the design of Tesla valves, which exhibits persistent vapor column with a duration of 34.5 ms in the inlet as a result of significant vapor backward flow[28], as illustrated in Supplementary Fig. 3. The suppression of vapor backward flow rendered by the Tesla valves is also reflected by the almost constant diodicity under different flow rates (Fig. 2c). Moreover, Fig. 2a, c indicate that the diodicity in two-phase regime is smaller than that of the single-phase regime at $Re > 2000$. This is because the two-phase flow (dominated by vapor-liquid-solid interface) exhibits a relatively small flow resistance as opposed to the single-phase flow that is associated with a larger Darcy friction factor (owing to its liquid-solid interface). Although directional two-phase flow can be achieved as a result of the presence of the Tesla valves, the Tesla channel is still limited by unstable and discontinuous flow (Supplementary Movie 2).

We demonstrate that, besides the suppression of vapor backflow via Tesla valves, achieving the thermal regulator with preferential thermal performances also requires the manifestation of a localized liquid spreading and the inhibition of the onset of unwanted dry-out area, an important feature enabled by the design of superhydrophilic pillars along the sidewalls of Tesla valves and main channel. Figure 2d shows the microscopic liquid rewetting process on both the Tesla valves and the superhydrophilic pillar arrays of the thermal regulator. We find that the introduction of superhydrophilic pillars transforms the originally unstable and discontinuous flow in Tesla channels into a stable and continuous profile in the thermal regulator, even in extreme boiling conditions, as demonstrated by the dynamic liquid rewetting in Fig. 2d and Supplementary Movie 3. Moreover, the enhanced capillary effect leads to a liquid renewal velocity of -0.5 m s⁻¹, which is much

faster than liquid spreading velocity of -1.8 mm s⁻¹ induced by capillary ratchets[17].

## Determination of the switching criterion for thermal regulator

After revealing the roles of Tesla valve and capillary structures on two-phase transport, we continue to characterize the thermal performance of our thermal regulator working in forward direction. Fundamentally, a high performance of convective boiling heat transfer demands efficient liquid-to-vapor transformation, defined as vapor quality ($\chi$). We first measured the time-dependent pressure drop and wall temperature behaviors by varying vapor quality. When $\chi < 0.25$, as shown in Fig. 3a, b, both pressure drop and wall temperature periodically fluctuate in a short period of -15 s, which is different from the random fluctuation observed in plain wall microchannels without Tesla valves[29]. In Supplementary Fig. 4, we observe pressure drop and wall temperature fluctuate synchronously. Except for thermal regulator, such similar periodic fluctuations are also manifested in Tesla channels (Supplementary Fig. 5). In the thermal regulator, the periodic oscillations are also accompanied with the vapor backflow near the inlet with a persistent duration, which is evidenced by the microscopic visualization of two-phase flow behaviors (Supplementary Fig. 6 and Supplementary Movie 4). The persistent vapor backflow blocks the liquid reflushing and results in the emergence of enlarged amplitudes in the pressure drop and wall temperature. In striking contrast, when $\chi > 0.25$, they become stabilized, showing a typical signature of a stable two-phase flow as a result of suppressing two-phase backflow owing to the utility of Tesla valves, termed as "switched-on" state, shown in Supplementary Fig. 7 and Supplementary Movie 4. Moreover, in switched-on state, $\chi$ directly regulates the pressure drop and wall temperature (Fig. 3c). Manifestation of such a switched-on

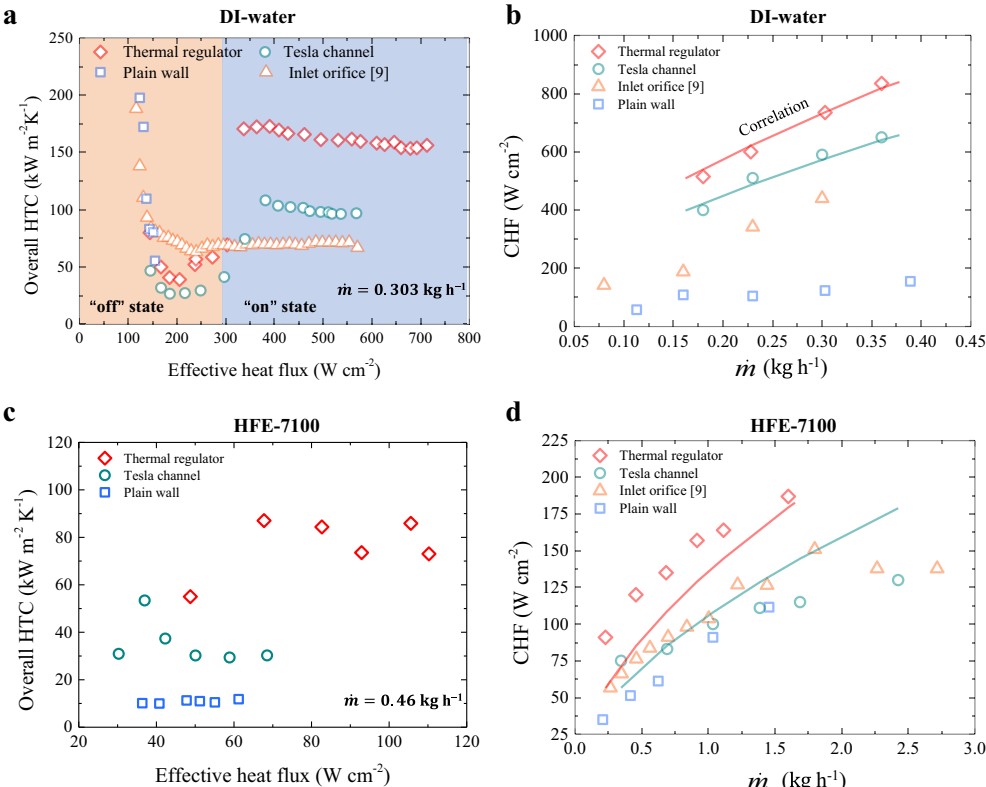

**Fig. 4 | Switchable and exceptional thermal performances of thermal regulator. a** Advantage of heat transfer performance of the thermal regulator over other control samples. The thermal regulator exhibits a switchable heat transfer performance, as reflected by the manifestation of an overall HTC of -175 kW m$^{-2}$ K$^{-1}$ in its switched-on state and a 5-fold drop in the switched-off state (in orange zone, $\chi < 0.25$, whereas in blue zone, $\chi > 0.25$). In contrast, the heat transfer of both plain wall and inlet orifice microchannels without the design of Tesla valves is non-switchable and also associated with a low overall HTC. Here, $\dot{m}$ is 0.303 kg h$^{-1}$. The error bars for overall HTC were -±6%. **b** Comparison of CHF between different microchannel configurations. Notably, a high CHF up to 830 W cm$^{-2}$ is achieved at $\dot{m} = 0.36$ kg h$^{-1}$, which is 5-fold larger than the plain wall configuration. CHF is predicted with a mean absolute error (MAE) of 17.6% using a correlation (see "Methods" section). The error bars of CHF were -±1%. **c** Notable enhancement of overall HTC for low surface tension fluid. A high overall HTC -80 kW m$^{-2}$ K$^{-1}$ is achieved, about 1.7-fold and 7-fold larger than that of the Tesla channel and plain wall, respectively. **d** The CHF of the thermal regulator compared to other three configurations. The CHF of the thermal regulator is reaching -200 W cm$^{-2}$ with 1-fold enhancement over others three configurations. Here $\dot{m}$ is 1.6 kg h$^{-1}$. The CHF of HFE-7100 is also nicely predicted with a MAE of 25%. Source data are provided as a Source Data file.

state upon reaching a critical $\chi$ of 0.25 is robust and also applicable to both Tesla channel and thermal regulator, as shown in Fig. 3d.

## Switchable and exceptional thermal performances

The switchable boiling heat transfer of the thermal regulator is closely dictated by its working states, which is different from tunable pool boiling heat transfer in opened space enabled by controlling nucleation "on" and "off" using external electric field[30] etc. As shown in Fig. 4a, the overall HTC curve displays a unique square root "√" shape, a nonlinear response to heat flux, demonstrating the switching ability. Supplementary Fig. 8 shows that there is a significant drop of wall superheat, which can be sustained at a relatively low value even at high heat fluxes when switching to "on-state". In contrast, it is difficult to control wall temperature using conventional designs. Thus, the switchability in flow boiling performance, which for the first time we have achieved, is transformative rather than incremental. In this work, we benchmark a switch ratio ($r = h_{on}/h_{off}$, where $h$ is heat transfer rate) about 6 via the design of thermal regulator. In the switched-off state, the overall HTC sharply declines to a low value after the onset of boiling. In contrast, switching on the thermal regulator when $\chi$ becomes higher than 0.25 leads to a sudden jump of the overall HTC up to -175 kW m$^{-2}$ K$^{-1}$ at $\dot{m} = 0.303$ kg h$^{-1}$. In striking contrast, the overall HTC of plain wall microchannels and inlet orifice sharply declines to about 50 kW m$^{-2}$ K$^{-1}$ and 75 kW m$^{-2}$ K$^{-1}$, respectively, after the onset of boiling. Thus, the overall HTC of thermal regulator is 2.4-

fold and 1.5-fold over the plain wall microchannels and typical microchannels with inlet orifices[9], respectively. Supplementary Fig. 9 shows the overall HTC curves under varying flow rates ranging from 0.23 kg h$^{-1}$ to 0.36 kg h$^{-1}$. Previously, Tesla-type valves that are capable of suppressing two-phase backflow have been largely used in oscillating/ pulsating heat pipes[30,31], however, they still suffer from a limited enhancement in heat transfer as imposed by low diodicity. By contrast, our thermal regulator can switch its working states in response to boiling conditions, leading to a giant boost in both high CHF and high overall HTC in its "switched-on" state. The advantage of our thermal regulator is shown in supplementary Table S1, which summarizes the typical switch ratios of different working mechanisms.

Our thermal regulator exhibits excellent thermal performances. Figure 4b shows that the CHF of the thermal regulator is ~830 W cm$^{-2}$ at $\dot{m} = 0.36$ kg h$^{-1}$, which remarkably outperforms that of Tesla channel, inlet orifice and plain wall. Especially, such thermal performance is 5-fold higher than the plain wall. Also, a high $\chi$ up to 0.8 at $\dot{m} = 0.18$ kg h$^{-1}$ is achieved, suggesting a super-efficient thermal transport of the thermal regulator (Supplementary Fig. 10). Moreover, the enhancement in the CHF and HTC of our thermal regulator is also achieved without sacrificing the $\Delta p$ (Supplementary Fig. 11). Table S2 presents a comprehensive comparison of normalized boiling heat transfer performances, and our thermal regulator exhibits much highest $q''_{CHF}/q''_{CHF-base}$ and $h_{HTC}/h_{HTC-base}$. Furthermore, a benchmark figure is presented in Fig. 5, which clearly indicates that the thermal

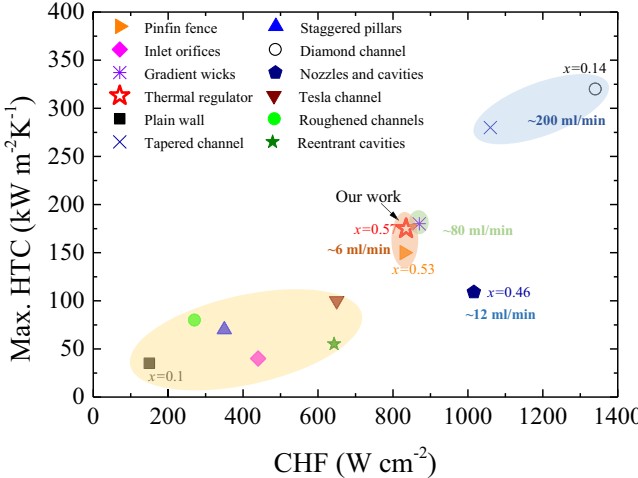

**Fig. 5 | A benchmark figure of flow boiling performances in parallel microchannels.** Yellow, orange, green, and blue zones represent four typical ranges of heat transfer rate. Table S3 summarizes the maximum HTCs at CHF conditions from different studies. Source data are provided as a Source Data file.

performance of our thermal regulator is excellent. It is noteworthy that the HTC of our work is the highest at the same order of magnitude of flow rates and channel hydraulic diameter ($D_h$). Furthermore, to predict CHF for DI-water and HFE-7100, we have successfully developed a new correlation by considering thermophysical properties of two liquid types, $D_h$ and channel length (L), which is written as:$q''_{CHF} = 0.18 G h_{fg} W e_{D_h}^{-0.2} (\frac{L}{D_h})^{0.57} (\frac{\rho_g}{\rho_l})^{1.13}$, where $W e_{D_h}$ is Weber number referred to channel hydraulic diameter ($G^2 D_h / \rho_l \sigma$), G is mass velocity, L is channel length, $D_h$ is hydraulic diameter, $\rho_l$ is liquid density, $\sigma$ is surface tension and $h_{fg}$ is latent heat of vaporization. The predictions of CHF in Fig. 4b nicely agree with experimental results with a mean absolute error (MAE) of 17.6% when using DI-water.

Finally, we extend our study by investigating the thermal regulator performances based on HFE-7100. Figure 4c shows that the overall HTC of the thermal regulator working in the forward direction is ~80 kW m$^{-2}$ K$^{-1}$, -1.7-fold and 7-fold larger than Tesla channel and plain wall microchannels, respectively. Additionally, the CHF value is approaching 200 W cm$^{-2}$ with 1-fold enhancement than others at ṁ = 1.6 kg h$^{-1}$ (Fig. 4d). The CHF of HFE-7100 is also well predicted with a MAE of 25% using the present model. Overall, the high performance of the thermal regulator on HFE-7100 suggests the efficacy and generality of fusing classical Tesla structures with new micropatterned superhydrophilic structures in reinforcing heat transfer performances. The concept of translating chaotic two-phase flow to directional flow and all the way to the switchable heat flow through physics-guided structural design would represent a paradigm in the thermal management of LED-light diode, high power laser and radar Transmit/Receive (T/R) modules and many other applications.

## Methods

### Rational design of the thermal regulator
Normally, due to the chaotic two-phase flows in confined spaces, boiling heat transfer performance is non-switchable on conventional designs. Particularly, a significant drop in heat transfer rate at high heat fluxes would result in unforeseen temperature rise in thermal devices. In our thermal regulator, we have rationally designed the Tesla valves with a high diodicity up to 4 in single-phase flow and decorated the sidewalls of both Tesla valves and main channel with regularly patterned pillar arrays to achieve a preferred and continuous liquid layer for effective thin film evaporation. The fusion of well-designed Tesla valves and pillar arrays is capable to achieve highly efficient two-phase transport and boost thermal performance. Specifically, the width of

Tesla bend and main channel, the dimensions of patterned pillar fences, and the ratio of Tesla bend width to main channel width are the main parameters that influence the thermal performance of our thermal regulator.

In this work, the diodicity is closely related to the pressure loss $\Delta p$ in Tesla bend, expressed as $\Delta p = \frac{1}{2} f_s \rho u^2 \frac{\pi R_b}{D} \frac{\theta}{180^\circ} + \frac{1}{2} k_b \rho u^2$, where $f_s$ is the Moody friction factor in a straight channel, $\rho$ is fluid density, u is the averaged flow velocity, D is hydraulic diameter of the Tesla bend, $R_b$ is bend radius, $\theta$ is bend angle, $k_b$ is bend loss coefficient. It is apparent that diodicity of Tesla valve is dependent on channel parameters, such as channel dimensions of D-hydraulic diameter of the Tesla bend, $R_b$-bend radius, and $\theta$-bend angle.

Furthermore, we have decoupled the effects of Tesla bend and Tesla island on boiling heat transfer in copper microchannels. Consequently, in Table S4, we find that Tesla bend can lead to an enhancement of CHF about 51% compared to plain wall microchannels. We further investigate the effect of Tesla island on thermal performance. The integration of Tesla island can further increase the CHF by 48%. On the other hand, we find that the ratio of width of Tesla bend to width of main channel would highly affect boiling performance in Tesla channels. Our work indicates that a ratio of about 0.5 can lead to much higher normalized boiling performance than the microchannel with Tesla valve (MCTV)[31], in which the ratio is -0.9. For example, at ṁ = 18.5 kg h$^{-1}$, the maximum heat flux is about 20 W cm$^{-2}$ using HFE-7100 in MCTV. In a striking contrast, a high value of 180 W/cm$^2$ is demonstrated at ṁ = 1.5 kg h$^{-1}$ in our thermal regulator, 9-fold higher than MCTV.

### Fabrication of thermal regulator
The thermal regulator was fabricated through deep reactive ion etching (DRIE). This thermal regulator consists of three parallel Tesla valves-based microchannels. The length of the tested chip is 10 mm. The critical dimension of width is 200 μm. The depth is 250 μm. The channel width of Tesla bend is 100 μm and the size ratio of Tesla bend to main channel is 0.5. Each channel includes 18 periodic Tesla valves. To boost the thin film evaporation, conformal micropillar arrays were fabricated close to the channel sidewalls. The diameter of pillar is 30 μm. The distance between two pillar is 10 μm. The gap between the sidewall of main channel and pillars is 15 μm. For comparison, we also fabricated two control samples, one of which consists of channels with Tesla valves but without the presence of pillar arrays and the other consists of plain-wall microchannels without the presence of typical Tesla valves.

We also designed a thin aluminum microheater deposited on backside of microdevice to simulate the heat source. This micro-heater functions as a micro-thermistor for measuring the average bottom surface temperature of the tested chip. The thickness from the heater to the channel bottom surface (channel base thickness) is 250 μm. Pyrex glass with thickness of 500 μm was used to anodically bond with the silicon substrate to form an integrated thermal regulator.

### Experimental measurements
Extensive experimental studies were conducted in this study using the setup illustrated in Supplementary Fig. 12. This setup mainly includes three key sub-systems, such as visualization study system comprised of high-speed camera (Phantom V 7.3) and Olympus microscope (BX-51) for, data acquisition system and fluid flows loop. DI-water and HFE-7100 were used to characterize thermal performance of our thermal regulator. Before experimental tests, degassed-processes were conducted to remove non-condensable gases. During experiments, the working fluids were driven through the tested microdevices using high-pressured nitrogen gas. The inlet/outlet pressures were collected using pressure transducers and then pressure drops crossing the tested microdevices can be deduced. To improve the measurement accuracy, flow meter Krohne Optimass 3300c was applied to the tests.

To characterize the heat transfer performances regarding HTC and CHF of our thermal regulator, we fabricated aluminum thin film-based micro-heater to simulate the working conditions of electronics. A power supply BK-PRECISION XLN10014 was applied to micro-heater and the voltage was measured by an Agilent digital multimeter (34972A). Uniform heat fluxes were generated on this micro-heater. Furthermore, after pre-calibration of this micro-heater, the average temperatures of microdevice were measured under various working conditions. Consequently, average wall temperatures of micro-channels were derived based on the temperatures of micro-heater. To assess the sensible heat of fluids, inlet/outlet fluid temperatures were measured. In this study, various parameters such as flow rate, pressure, temperature, voltage and current were monitored and collected by a customized data acquisition system. The dynamic two-phase flows in confined space were recorded using the visualization system. All experiments were conducted at 1 atm.

## Correlation of CHF

A new CHF correlation is developed by considering thermophysical properties of DI-water and HFE-7100, channel hydraulic diameter ($D_h$) and channel length ($L$). Previously, a non-dimensional correlation for CHF was proposed by Hall and Mudawar[32], which is expressed by

$$Bo_{CHF} = f\left(We, \frac{\rho_l}{\rho_g}, \chi_{e,in}, \frac{L}{D_h}\right) \tag{1}$$

In this work, we developed a new CHF correlation in a form similar to the reported CHF correlation by Mudawar et al.[33] that is

$$q''_{CHF} = C_1 We_{D_h}^{C_2}\left(\frac{L}{D_h}\right)^{C3}\left(\frac{\rho_f}{\rho_g}\right)^{C4} Gh_{fg} \tag{2}$$

where $We_{D_h}$ is Weber number based on $\frac{G^2 D_h}{\rho_l \sigma}$, $G$ is the mass velocity, $L$ is the channel length, $D_h$ is the hydraulic diameter of channel, $\rho_l$ is the liquid density, $\sigma$ is the surface tension, $h_{fg}$ is latent heat of vaporization. C1-C4 are 0.18, −0.2, 0.57, and 1.13, respectively. Finally, this correlation can nicely predict CHF with MAE of 17.6% and 25% for DI-water and HFE-7100, respectively.

## Data availability

The data are provided in this article and Supplementary Information. Source data are provided with this paper.

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

## Acknowledgements

We acknowledge financial support from the National Natural Science Foundation of China under the grants U2241253 to Dr. Yongping Chen as well as General Research Fund (B-QC0R) and RGC Senior Research Fellow Scheme (3-RA87) to Dr. Zuankai Wang.

## Author contributions

Y.C., W.L., and Z.W. conceived the research. W.L., Z.W., Y.C., and C.L. designed the research. W.L. and S.Y. carried out the experiments. W.L. and S.Y. conducted data and image analysis. All the authors contributed to data reduction and data analysis. Z.W., Y.C., W.L., S.Y. and C.L. wrote the manuscript and all the authors agreed on its final contents.

## Competing interests

The authors declare no competing interests.
