## [Peer Review File · Nature Communications]

Tesla valves and capillary structures-activated thermal regulatorEditorial Note: This manuscript has been previously reviewed at another journal that is not operating a transparent peer review scheme. This document only contains reviewer comments and rebuttal letters for versions considered at *Nature Communications*.

REVIEWERS' COMMENTS

Reviewer #1 (Remarks to the Author):

I continue to appreciate the authors efforts to clarify their text in response to the reviewer comments. The benchmark figure (Fig. R6) provides a nice visual to contextualize the results, and seems like it would be more appropriate include in the main text either as a first or last figure. In fact doing so could certainly help the paper to be more approachable to a broader audience, which would help justify its inclusion in a broad-audience journal.

It's unfortunate that the authors are unable to provide the physical insight that could come from first-principle modeling. I'd strongly encourage the authors to replace the word "model" with "correlation" in the final paragraph on Figure 4b. It might also be appropriate to identify any clear limits to this correlation. For example, if the channel length L were quadrupled or the mass velocity G were tripled, do the authors anticipate that the critical heat flux would exceed that of all of the other devices shown in the benchmark figure (eg $>1400 \text{ W/cm}^2$)?

As a minor suggestions, the use of novel acronyms, such as TC for Tesla channels, is unnecessary and makes the paper harder to follow. I would suggest removing these acronyms whenever possible.

Reviewer #2 (Remarks to the Author):

The author has addressed all my previous questions. Even though the manuscript is lack of fundamental modeling based on first principles, I believe that the part of fabrication and characterizations of this novel structure is still strong.

Point-to-point responses to referees' comments

Referee #1 (Remarks to the Author):

1. I continue to appreciate the authors efforts to clarify their text in response to the reviewer comments. The benchmark figure (Fig. R6) provides a nice visual to contextualize the results, and seems like it would be more appropriate include in the main text either as a first or last figure. In fact doing so could certainly help the paper to be more approachable to a broader audience, which would help justify its inclusion in a broad-audience journal.

Response: We are pleased that the referee appreciates our effort to improve the manuscript. Following referee's suggestion, we have added the benchmark figure to the main text as Fig. 5.

2. It's unfortunate that the authors are unable to provide the physical insight that could come from first-principle modeling. I'd strongly encourage the authors to replace the word "model" with "correlation" in the final paragraph on Figure 4b.

Response: We have replaced "model" with "correlation" on Fig. 4b.

3. It might also be appropriate to identify any clear limits to this correlation. For example, if the channel length L were quadrupled or the mass velocity G were tripled, do the authors anticipate that the critical heat flux would exceed that of all of the other devices shown in the benchmark figure (eg >1400 W/cm²)?

Response: We thank the referee for the insightful comments. In our correlation, the mass velocity G , the channel length L and the hydraulic diameter of channel D_h are the main factors that determine the value of critical heat flux. Therefore, the critical heat flux might exceed that of all of the other devices shown in the benchmark figure (eg >1400 W/cm²) by optimizing G , L and D_h , etc.

4. As a minor suggestions, the use of novel acronyms, such as TC for Tesla channels, is unnecessary and makes the paper harder to follow. I would suggest removing these acronyms whenever possible.

Response: Following the referee's suggestion, we have changed TC to Tesla channel in the main text and on the figures.

Reviewer #2 (Remarks to the Author):

The author has addressed all my previous questions. Even though the manuscript is lack of fundamental

modeling based on first principles, I believe that the part of fabrication and characterizations of this novel structure is still strong.

Response: We thank the referee for the time spent on our paper and complimentary comments. We are pleased that the referee appreciates our effort to improve the manuscript.